# Cetuximab for Immunotherapy-Refractory/Ineligible Cutaneous Squamous Cell Carcinoma

**DOI:** 10.3390/cancers15123180

**Published:** 2023-06-14

**Authors:** Julian A. Marin-Acevedo, Bethany M. Withycombe, Youngchul Kim, Andrew S. Brohl, Zeynep Eroglu, Joseph Markowitz, Ahmad A. Tarhini, Kenneth Y. Tsai, Nikhil I. Khushalani

**Affiliations:** 1Medical Oncology, Indiana University Melvin and Bren Simon Comprehensive Cancer Center, Indianapolis, IN 46202, USA; 2Department of Cutaneous Oncology, Moffitt Cancer Center, Tampa, FL 33612, USA; bethany.withycombe@moffitt.org (B.M.W.); andrew.brohl@moffitt.org (A.S.B.); zeynep.eroglu@moffitt.org (Z.E.); joseph.markowitz@moffitt.org (J.M.); ahmad.tarhini@moffitt.org (A.A.T.); 3Department of Biostatistics and Bioinformatics, Moffitt Cancer Center, Tampa, FL 33612, USA; youngchul.kim@moffitt.org; 4Department of Pathology, Division of Dermatopathology, Moffitt Cancer Center, Tampa, FL 33612, USA; kenneth.tsai@moffitt.org

**Keywords:** cutaneous squamous cell carcinoma, immunodeficiency, therapeutics, cetuximab, immune checkpoint inhibitor, immunotherapy

## Abstract

**Simple Summary:**

Cetuximab remains a viable treatment for patients with advanced cutaneous squamous cell carcinoma who fail or are ineligible for immunotherapy. When used immediately following the progression of anti-PD1 therapy, cetuximab demonstrated particularly rapid and durable responses in our single institution retrospective experience. These findings should be validated in larger, prospective studies. However, if results are confirmed, cetuximab should be considered the preferred second-line agent after the failure of ICI. Future research should explore how ICI impacts subsequent anti-EGFR therapy, determine the ideal sequencing strategy when these agents are used and define if combination therapy using ICI with cetuximab is better than using either agent alone.

**Abstract:**

Anti-PD1 therapy demonstrated impressive, prolonged responses in advanced cutaneous squamous cell carcinoma (CSCC). Therapy for ICI-refractory/ineligible disease remains unclear. We performed a retrospective analysis in locally-advanced/metastatic CSCC using cetuximab across three cohorts: immediately after ICI failure (A), not immediately following ICI failure (B), or without prior ICI (C). The primary endpoint was the overall response rate (ORR). Secondary endpoints included disease-control rate (DCR), progression-free survival (PFS), overall survival (OS), time-to-response (TTR) and toxicity. Twenty-three patients were included. In cohort A (n = 11), the ORR was 64% and DCR was 91%, with six ongoing responses at data cutoff. In cohort B (n = 2), all patients had progression as the best response. At a median follow-up of 21 months for A and B, TTR and PFS were 2.0 and 17.3 months, respectively. The median OS was not reached. In cohort C (n = 10), the ORR and DCR were 80%, including five ongoing responses at the data cutoff. At a median follow-up of 22.4 months, the TTR, PFS and OS were 2.5, 7.3 and 23.1 months, respectively. Cetuximab was well tolerated in all cohorts. In summary, cetuximab is effective in patients with failure/contraindications to ICI. Cetuximab immediately after ICI failure yielded particularly fast, durable responses. If confirmed, this could be the preferred therapy following ICI failure.

## 1. Introduction

Cutaneous squamous cell carcinoma (CSCC) used to be the second most common keratinocyte carcinoma after basal cell carcinoma, but recent findings suggest they are now equally common, occurring at a 1:1 ratio [1,2]. The reported incidence has dramatically increased up to 200% in the past decades and it is expected to continue its rising trend [1,3]. Most cases of CSCC are diagnosed at an early stage and are curable with surgery or other ablative modalities. However, a small number of patients have CSCC with high-risk features that portend the risk for regional and distant metastases. In a retrospective cohort study of 985 CSCC patients with 1832 tumors, the risk for regional nodal metastases was 3.7% and the risk of disease-specific mortality was 2.1% [4]. Specific clinicopathologic factors for high-risk disease from this and other studies include tumor thickness (>4 mm), invasion beyond the subcutaneous fat, perineural invasion, diameter (≥2 cm), desmoplasia, poorly differentiated histology, anatomic location (ear, temple, or lips) and immunosuppression [3,4,5]. Historically, the treatment of unresectable or metastatic CSCC has been of limited efficacy. Platinum-based chemotherapy regimens have shown variable response rates with limited duration of response (DOR) and an often limiting toxicity profile [6,7,8,9,10,11]. Cetuximab is a monoclonal antibody targeting the epidermal growth factor receptor (EGFR), which is typically overexpressed in CSCC [11]. It is thought to be better tolerated than chemotherapy, particularly in elderly patients who comprise the majority of cases of advanced CSCC [12]. Cetuximab was evaluated in a prospective phase II clinical trial in chemotherapy-naïve advanced CSCC patients with moderate or strong EGFR expression by immunohistochemistry [13]. Cetuximab monotherapy resulted in an overall response rate (ORR) of 28% and a disease control rate (DCR) at 6 weeks of 69% [13]. For patients with an objective response, the median DOR was 6.8 months. The median progression-free survival (PFS) and overall survival (OS) were 4.1 months and 8.1 months, respectively [13]. Cetuximab had an acceptable safety profile with an acneiform rash being the most common toxicity [13]. A more recent retrospective study using frontline cetuximab in 58 patients with unresectable CSCC demonstrated a 6-week DCR of 87%, an ORR of 53%, a median PFS of 9.7 months and a median OS of 17.5 months [14]. The improved outcomes seen in this study were attributed to the lower prevalence of advanced disease and lymph node involvement compared to the prospective clinical trial population [14]. 

The therapeutic landscape of advanced CSCC has dramatically changed with the advent of immune checkpoint inhibitor (ICI) therapy. Cemiplimab, an anti-programmed cell death protein-1 (PD-1) agent, demonstrated excellent efficacy in patients with locally advanced or metastatic CSCC in a phase I/II clinical trial [15,16]. In this setting, cemiplimab demonstrated a DCR of 65–79%, an ORR of 44–47%, a median time to response (TTR) of 1.9 months, a median PFS of 18.4–21.7 months, a DOR between 41.3 months and not reached and an OS between 48.4 months and not reached [15,16,17]. Based on these results, cemiplimab was the first ICI to be approved for advanced CSCC [18]. Pembrolizumab, an anti-PD1 agent, was also investigated in patients with locally advanced or recurrent/metastatic CSCC [19]. It demonstrated a DCR of 52–65%, an ORR of 35–50%, a median TTR of 2.6 months, a median PFS of 5.7 months, a median OS of 23.8 months and a median DOR that was not reached [19]. With these results, pembrolizumab was also approved as a first-line option for advanced CSCC [20].

Despite the excellent therapeutic benefit of ICI, many patients experience primary or secondary resistance to treatment [21]. In addition, many patients may not be eligible for ICI therapy due to pre-existing auto-immune disease or the need for immunosuppression [22]. Therefore, the optimal therapeutic strategy for these situations remains unknown and represents an unmet need in advanced CSCC. There is no standard second-line therapy following the failure of anti-PD1 treatment in advanced CSCC and consideration for clinical trials is very appropriate. Our anecdotal experience using cetuximab following anti-PD1 failure was encouraging and became our preferred approach in the management of these patients. We report our institutional experience using cetuximab after ICI failure as well as its role among CSCC patients with locally advanced and metastatic disease in whom upfront ICI was deemed inappropriate.

## 2. Patients and Methods

A single-institution retrospective review was conducted from 28 September 2018 (the date of cemiplimab FDA approval for CSCC) with mature follow-up data through 30 April 2022. Patients with locally advanced or metastatic CSCC who received cetuximab for this diagnosis at the Moffitt Cancer Center were included in the study. These patients were divided into three cohorts: those who received cetuximab immediately following progression on ICI (cohort A), as a subsequent line with alternate intervening therapy following progression on ICI therapy (cohort B), or those who received cetuximab without prior ICI (cohort C). The primary endpoint was ORR. Secondary endpoints included DCR, TTR, PFS, OS and toxicity (Figure 1). Descriptive analyses were performed using the median and ranges for continuous variables and proportions and frequencies for categorical variables. Toxicities were graded based on the Common Terminology Criteria for Adverse Events (CTCAE) version 5.0 [23]. Responses were recorded using the response evaluation criteria in solid tumors (RECIST) version 1.1 into complete response (CR), partial response (PR), stable disease (SD) and progressive disease (PD) categories [24]. The ORR was defined as the percentage of patients with CR and PR, while DCR was defined as the percentage of patients with CR, PR and SD. Survival data were calculated using the Kaplan–Meier method. R version 4.1.0 was used for statistical data analysis. This study was conducted with appropriate regulatory measures, performed in accordance with the Helsinki Declaration and comparable ethical standards and approved by the institutional review board (IRB) at the University of South Florida. Considering that all the procedures performed were part of routine care and the retrospective nature of the analysis, informed consent and ethical approval was waived by the IRB of the University.

## 3. Results

### 3.1. Patients

A total of 25 patients were identified during the timeframe of the study. Of these, 2 patients were excluded from the analysis: 1 patient had a diagnosis of cutaneous sarcoma after a pathology review and the other patient developed a severe infusion reaction during the first administered dose of cetuximab and did not receive additional doses. The remaining 23 patients were analyzed. Table 1 summarizes the demographics and baseline characteristics of the study population. Thirteen patients received cetuximab after failing ICI (cohorts A and B). Of these, 11 (85%) received cetuximab immediately following ICI progression (cohort A), including 6 (46%) who also received concurrent palliative or definitive radiation (Table 2). Two patients (15%) received additional intervening therapy following ICI failure and prior to receiving cetuximab (cohort B). These included chemotherapy with carboplatin plus paclitaxel in one patient and a clinical trial therapy for the second patient. Neither patient in cohort B received concurrent radiotherapy. Cohort C comprised 10 patients who received cetuximab without prior use of ICI. Three of these patients received cetuximab with concurrent definitive radiation (Table 2).

In cohorts A and B, the median age was 72 years. The majority (n = 11; 85%) were males and all patients were white. One patient in cohort B had concurrent chronic lymphocytic leukemia (CLL) managed with obinutuzumab upon initiation of systemic therapy for CSCC. Primary sites of disease included the extremities (n = 6; 46%), face (n = 3; 23%), scalp (n = 2; 15%) and trunk (n = 2; 15%). Only three patients (23%) had locally advanced/unresectable disease while the remaining (n = 10; 77%) had metastatic disease. Sites of metastasis included the lymph nodes (n = 5), lungs (n = 3), skin (n = 1) and central nervous system (CNS) (n = 1).

In cohort C, the median age was 68 years. Most patients (n = 6; 60%,) were females and all of them were white. Most patients (n = 8; 80%) were receiving concurrent immunosuppressive therapies to prevent graft rejection from prior solid organ transplantation (50%, n = 5) or to treat some form of severe autoimmune condition (n = 3; 30%,) (Table 3). The two patients who were not on immunosuppressive therapy had either a well-controlled autoimmune disorder or had opted for concurrent chemo-radiation over immunotherapy. Primary sites of disease included the face (n = 7; 70%), extremities (n = 2; 20%) and trunk (n = 1; 10%). Most patients (n = 6; 60%) had locally advanced/unresectable disease, while all the remaining patients (n = 4; 40%) had metastatic disease to the lymph nodes. Most patients (n = 9, 90%), received cetuximab as their first systemic treatment, including one patient who had received intralesional therapy (with 5-FU and bleomycin) and another patient who had completed definitive radiation. Only one patient, with a history of heart transplantation, received cetuximab as second-line after carboplatin and paclitaxel.

### 3.2. Efficacy

Table 4 summarizes the best responses to cetuximab in all cohorts. The median TTR in cohorts A and B was 2 months (0–9 months, 95% CI). In cohort A, the ORR was 64% and the DCR was 91% including one patient with CR, six with PR, three with SD and one with PD. All patients who received concurrent cetuximab with radiation had either a PR (n = 4/6) or SD (n = 2/3). These responses, however, were not deemed to be the sole result of radiation therapy given that all of them, except for one patient with SD, had measurable disease outside of the radiation field (Table 2). Six of the patients with either a CR (n = 1/1), PR (n = 4/6), or SD (n = 1/3) had ongoing responses at the time of data cutoff. All of them had discontinued cetuximab due to the patient/physician’s choice. The other four patients with either PR (n = 2/6) or SD (n = 2/3), progressed during subsequent follow-up. This included one patient who achieved a PR and had been off cetuximab for almost 16 months prior to progression. In those who progressed after cetuximab use (one upfront and four after PR or SD), three received subsequent chemotherapy with carboplatin and paclitaxel, one received an experimental agent and one did not receive any subsequent therapies. Those patients who received subsequent chemotherapy had a PR (n = 1), SD (n = 1), or died from unclear reasons prior to response evaluation (n = 1). The patient on the experimental agent achieved SD as the best response.

In cohort B, the best response to cetuximab was PD (n = 2). None of these patients had received concurrent radiation. At progression, subsequent systemic therapies included carboplatin with paclitaxel with PR to therapy and ipilimumab with nivolumab with PD to therapy.

In cohort C, the median TTR was 2.5 months (0–6 months, 95% CI). The ORR and DCR were both 80%, including one CR, seven PR and two PD. All patients who received concurrent radiation (n = 3) achieved a PR (Table 2). One patient with CR and two with PR had ongoing responses at the time of data cutoff and had discontinued cetuximab due to toxicity or physician’s choice after one year of therapy. One patient who died from a multiple sclerosis flare attributed to cetuximab had an ongoing PR at the time of demise. Another patient with PR was lost to follow-up. The remaining three patients with PR eventually progressed, including one patient who had been off therapy for seven months after receiving cetuximab for one month and stopping it due to toxicity. Among those who progressed (two upfront and three after PR), one did not receive any additional systemic therapy, three received subsequent carboplatin and paclitaxel (achieving PR followed by PD, SD, or PD) and one patient was managed with an experimental drug but was transitioned promptly to cemiplimab due to poor tolerance. Both patients who progressed on chemotherapy (both kidney transplants) went to receive subsequent nivolumab or cemiplimab. Interestingly, all three patients treated with an ICI had PD as the best response.

At the time of data cutoff and with a median follow-up duration of 21 months, the median PFS was 17.3 months (5.25–NR; 95% CI) and the median OS had not been reached for cohorts A and B (19.35–NR; 95% CI) (Figure 2). Three patients had died due to PD, including two patients from cohort A who had achieved a PR or SD and had received concurrent radiation. The other patient was in Cohort B, had a history of CLL and had achieved PD as the best response to both cetuximab and ipilimumab with nivolumab.

With a median follow-up of 22.4 months, the median PFS was 7.3 months (2.64–NR; 95% CI) and the median OS was 23.1 months (8.5–NR; 95% CI) for cohort C (Figure 2). Five patients had died, three from PD, one from a multiple sclerosis flare and one from COVID-19 infection.

### 3.3. Toxicities

Treatment-related toxicities were seen in 96% of all patients, particularly rash (70%) and hypomagnesemia (61%) (Table 5). Most toxicities were grade 1 or grade 2.

Among cohorts A and B, all patients developed at least one toxicity but none of these led to therapy discontinuation. In cohort C, nine patients (90%) developed at least one therapy-related toxicity, including four in whom cetuximab was discontinued. This included one patient with grade 2 rash, one patient with grade 2 hypomagnesemia, one with grade 2 rash plus hypomagnesemia and one patient with a grade 5 toxicity consisting of a suspected fatal multiple sclerosis flare that was attributed to cetuximab by the treating provider.

## 4. Discussion

In this retrospective analysis, we demonstrate that cetuximab remains an effective therapeutic option for patients with advanced CSCC who have failed or who are not ideal candidates for ICI in the front-line setting.

It is difficult to compare our results with other analyses performed in the pre-ICI era. However, the use of cetuximab after anti-PD1 failure seems to be associated with better outcomes than seen in a previous prospective trial using frontline cetuximab [13]. We noticed a better ORR (54% versus 28%), DCR (77% versus 69%), median PFS (17.3 versus 4.1 months) and median OS (not reached versus 8.1 months) [13]. A recent analysis of a significantly smaller population also demonstrated a high ORR with the use of cetuximab after IO failure consistent with our findings [25]. Responses in our cohort were fast, with a median TTR of two months. Most patients were males and had no significant comorbidities, autoimmune conditions, or history of solid organ transplantation, except for a single patient with CLL on obinutuzumab. All responses were seen in those exposed to cetuximab immediately after ICI failure. In this subgroup, the ORR was approximately 64% and the DCR was 91%. Those who received other forms of therapy between ICI and cetuximab did not demonstrate any response in our series.

Responses among those who received cetuximab without prior ICI exposure were fast (median 2.5 months) and somewhat similar to those seen in a previous retrospective analysis of cetuximab from the pre-ICI era [14]. The PFS and OS in our cohort, however, were more promising than reported in the clinical trial by Maubec et al. (PFS of 4.1 vs. 7.3 months and OS of 8.1 vs. 23.1 months) [13]. These discrepancies may reflect the differences in patients’ demographics. While in the clinical trial, most patients were males, none were immunosuppressed and all patients had EGFR overexpression [13], our patients in cohort C were predominantly females and most had either an autoimmune disorder (40%) or a history of solid organ transplantation (50%) (80% were on systemic immunosuppressive therapy) and we did not specifically select for CSCC with EGFR overexpression. Cohort C also differed substantially from cohorts A and B; only three patients (two of them on immunosuppression for kidney transplantation and one with no immunosuppressive therapy) received ICI as a later-line therapy after cetuximab. Interestingly, all of them had PD as their best response to ICI.

Overall, cetuximab was well tolerated in each cohort and had a safety profile consistent with previous studies [26]. We did not find any substantial differences in the toxicity profile when cetuximab was used before or after ICI, suggesting that prior ICI does not affect the toxicity profile of cetuximab. However, there was a higher incidence of cetuximab discontinuation due to treatment-related toxicities among those without prior ICI.

To the best of our knowledge, this is the first report demonstrating the value of cetuximab in patients who have failed or who are not candidates for upfront anti-PD1 therapy in the modern era. Most data available for cetuximab in CSCC is in the first-line setting, prior to ICI becoming the standard of care for patients with advanced disease [12,13,14]. The few reports on second-line agents for advanced CSCC following the failure of non-ICI systemic therapy options have mainly used carboplatin with paclitaxel, cetuximab, or docetaxel and the median duration of therapy was only 3.4 months [27]. Recently, the addition of cetuximab to pembrolizumab demonstrated an ORR of 44% among those with primary or acquired resistance to ICI in advanced CSCC [28]. Combination therapy, however, resulted in a higher incidence of grades 3–4 AEs (35%) and it is unclear if the responses seen were a result of combination therapy or cetuximab itself [28]. While an ongoing clinical trial is assessing the role of anti-EGFR therapy using afatinib in patients who have failed immunotherapy (NCT05070403), we believe a clinical trial should explore the role of cetuximab after ICI failure. Other anti-EGFR monoclonal antibodies (i.e., panitumumab) or tyrosine kinase inhibitors (i.e., gefitinib, erlotinib, afatinib, dacomitinib, or osimertinib) could also be explored.

Intriguing findings from our study include the substantial differences in outcomes seen among patients who received cetuximab immediately after ICI compared to those who received other forms of therapy in between ICI and cetuximab. In addition, responses among those patients who received ICI after being exposed to cetuximab were generally poor. The reason for these findings is unclear. It is known that the EGFR signaling pathway modulates the tumor microenvironment and that anti-EGFR therapy directly affects responses to ICI [29,30]. Based on this theory, one may speculate that the use of cetuximab prior to ICI could have negatively impacted responses to anti-PD1 and anti-CTLA-4 agents in our patients. Additionally, one may hypothesize that the reverse condition also stands and that ICI prior to cetuximab could modulate the tumor microenvironment and EGFR pathway, leading to improved responses to anti-EGFR therapy. Based on this hypothesis and the clinical results seen with other malignancies such as melanoma [31], we believe the use of upfront ICI, when feasible, followed by targeted therapy at progression, could result in better outcomes compared to the opposite approach.

The major limitations of our study include its single-institution retrospective design and the small population included. The generalizability of our findings is limited by the fact that only white individuals were represented. Since we did not purposely exclude individuals of different races/ethnicities, this likely represents that CSCC is more common among white patients [3]. A substantial proportion of patients (~40%) received palliative or definitive radiation in conjunction with the cetuximab which could have confounded the results. However, given that most patients had evaluable responses outside of the radiated field, that all CRs were seen in patients who did not receive radiation and that some of those with ongoing responses at the time of data cutoff had not received radiation, it is our thought that cetuximab alone has activity irrespective of radiation. Finally, the discrepancies seen between cohorts A and B are limited by the small number of patients in cohort B.

## 5. Conclusions

Cetuximab remains an effective treatment in the post-ICI era. For patients without contraindications to ICI, the use of cetuximab immediately after progression on anti-PD1 therapy demonstrated particularly fast and long-lasting responses. This should be validated in larger datasets and prospective studies. If results are confirmed, then cetuximab should be considered the preferred second-line agent after the failure of ICI. For those with contraindications to ICI, cetuximab remains an effective frontline therapy. Future research should explore how ICI impacts anti-EGFR therapy, determine the ideal sequencing strategy when these agents are used and define if combination therapy using ICI with cetuximab is better than using either agent alone.

## Figures and Tables

**Figure 1 cancers-15-03180-f001:**
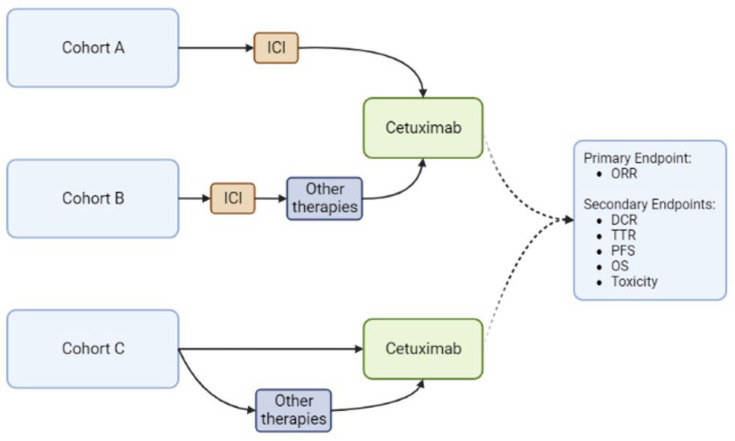
Study Design. DCR (Disease Control Rate), ORR (Overall Response Rate), OS (Overall Survival), PFS (Progression-Free Survival), TTR (Time to Response). Created with BioRender.com.

**Figure 2 cancers-15-03180-f002:**
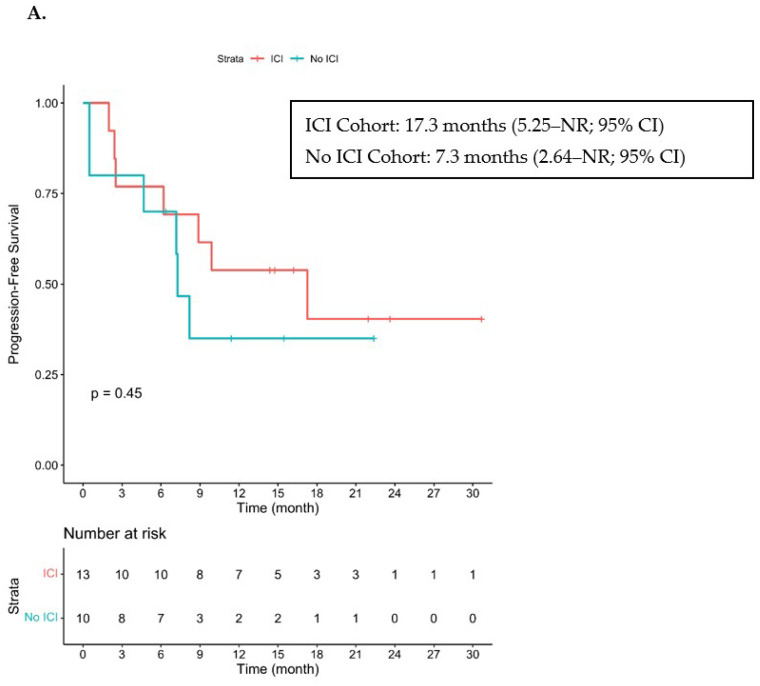
Kaplan–Meier Curve of PFS and OS. ICI—immune checkpoint inhibitor. (**A**) Progression-free survival in patients who received ICI prior to cetuximab (cohorts A + B) and in those who did not receive ICI prior to cetuximab (cohort C). (**B**) Overall survival in patients who received ICI prior to cetuximab (cohorts A + B) and in those who did not receive ICI prior to cetuximab. NR stands for Not Reached.

**Table 1 cancers-15-03180-t001:** Demographics of patients treated with cetuximab.

	All Patients (N = 23)	Cohorts A and B (N = 13)	Cohort A (N = 11)	Cohort B (N = 2)	Cohort C (N = 10)
**Age**	68 {49–89}	72 {54–89}	72 {54–89}	71.5 {65–77}	67.5 {49–75}
**Gender**					
Males	15 (65.2%)	11 (84.6%)	10 (90.9%)	1 (50%)	4 (40%)
Females	8 (34.8%)	2 (15.4%)	1 (9.1%)	1 (50%)	6 (60%)
**Race**					
White	23 (100%)	13 (100%)	11 (100%)	2 (100%)	10 (100%)
**Comorbidities on Systemic Immunosuppression or Lymphodepleting Therapy**	9 (39.1%)	1 (7.7%)	-	1 (50%)	8 (80%)
Secondary Cancer	1 (4.3%)	1 (7.7%)—CLL	-	1 (50%)—CLL	-
Autoimmunity	3 (13%)	-	-	-	3 (30%)
Kidney Transplant	3 (13%)	-	-	-	3 (30%)
Heart Transplant	1 (4.3%)	-	-	-	1 (10%)
Liver Transplant	1 (4.3%)	-	-	-	1 (10%)
**Primary Site**					
Scalp	2 (8.7%)	2 (15.4%)	2 (18.2%)	-	-
Face	10 (43.5%)	3 (23.1%)	3 (27.3%)	-	7 (70%)
Trunk	3 (13%)	2 (15.4%)	1 (9.1%)	1 (50%)	1 (10%)
Upper Extremity	3 (13%)	3 (23.1%)	2 (18.2%)	1 (50%)	-
Lower Extremity	5 (21.7%)	3 (23.1%)	3 (27.3%)	-	2 (20%)
**Disease Stage at Time of Therapy**					
Locally Advanced	9 (39.1%)	3 (23.1%)	3 (27.3%)	-	6 (60%)
Metastatic	14 (60.9%)	10 (76.9%)	8 (72.3%)	2 (100%)	4 (40%)
**Line Cetuximab Use**					
First	9 (39.1%)	-	-	-	9 (90%)
Second or More	14 (60.9%)	13 (100%)	11 (100%)	2 (100%)	1 (10%)
**Concurrent Radiation**	9 (39.1%)	6 (46.2%)	6 (54.5%)	-	3 (30%)

CLL—chronic lymphocytic leukemia.

**Table 2 cancers-15-03180-t002:** Concurrent Radiation dosing and location.

Cohort	Patient	Location of Radiation	Dosing and Fractions	Radiation Goal	Measurable Distant Disease	Response to Cetuximab	Progression after Cetuximab
A (n = 6/11, 54.5%)	1	Primary Tumor	66 Gy, 33 fractions	Definitive	Yes: Distal Lymph Nodes	PR	No
	2	Regional Lymph Node	30 Gy, 10 fractions	Palliative	Yes: Distal Lymph Nodes	PR	No
	3	Regional Lymph Node	54 Gy, 18 fractions	Palliative	Yes: Distal Lymph Nodes and Bone	SD	Yes
	4	Primary Tumor + Regional Lymph Node	70 Gy, 35 fractions	Definitive	No	SD	No
	5	Metastatic Cutaneous Lesion	40 Gy, 10 fractions	Palliative	Yes: Distal Lymph Nodes	PR	Yes
	6	Primary Tumor	66 Gy, 33 fractions	Definitive	Yes: Pulmonary Metastasis	PR	No
C (n = 3/10, 30%)	1	Primary Tumor	70 Gy, 35 fractions	Definitive	No	PR	No
	2	Regional Lymph Nodes	30 Gy, 15 fractions	Palliative	No	PR	Yes
	3	Regional Lymph Nodes	70 Gy, 35 fractions	Definitive	No	PR	Yes

Gy—gray, PR—partial response and SD—stable disease.

**Table 3 cancers-15-03180-t003:** Concurrent immunosuppressive therapies and best response to cetuximab (Cohort C).

Associated Condition (n = 10)	Immunosuppressive Regimens	Concurrent Radiation	Best Response	Progression after Cetuximab
Kidney Transplantation (n = 3)	Cyclosporine + prednisone (n = 1)	No	CR	No *
	Tacrolimus (n = 1)	No	PD	N/A
	Tacrolimus + mycophenolate (n = 1)	No	PD	N/A
Liver Transplantation (n = 1)	Tacrolimus + mycophenolate	Yes	PR	No
Heart Transplantation (n = 1)	Tacrolimus + prednisone	Yes	PR	Yes
Multiple Sclerosis (n = 1)	Fingolimod	No	PR	No *
Wegener’s Granulomatosis (n = 1)	Prednisone	No	PR	Yes
Severe Lichen Planus (n = 1)	Beclomethasone + triamcinolone	No	PR	No *
Sjögren’s Syndrome (n = 1)	None (n = 1)	No	PR	No **
None (n = 1)	None (n = 1)	Yes	PR	Yes ***

* Discontinued cetuximab due to toxicity at the time of cut-off. ** Discontinued cetuximab per patient preference. ******* After being off cetuximab for seven months.

**Table 4 cancers-15-03180-t004:** Best response to cetuximab in Cohorts A, B and C.

Response	All Patients (N = 23)	Cohorts A/B (N = 13)	Cohort A (N = 11)	Cohort B (N = 2)	Cohort C (N = 10)
ORR	65.2% (15)	53.9% (7)	63.6% (7)	0% (0)	80% (8)
DCR	78.2% (18)	77% (10)	90.9% (10)	0% (0)	80% (8)
CR	8.7% (2)	7.7% (1)	9.1% (1)	-	10% (1)
PR	56.5% (13)	46.2% (6)	54.5% (6) *	-	70% (7) ****
SD	13% (3)	23.1% (3)	27.3% (3) **	-	-
PD	21.7% (5)	23.1% (3)	9.1% (1)	100% (2) ***	20% (2)

CR—complete response, DCR—disease control rate, ORR—overall response rate, PD—progressive disease, PR—partial response and SD—stable disease. * Four patients received concurrent radiation therapy. ** Two patients received concurrent radiation therapy. *** One patient was on obinutuzumab for chronic lymphocytic leukemia. **** Three patients received concurrent radiation therapy.

**Table 5 cancers-15-03180-t005:** Main toxicities attributed to cetuximab.

Toxicities	All Patients (N = 23)	Cohort A (N = 11)	Cohort B (N = 2)	Cohort C (N = 10)
All	22 (95.7%)	11 (100%)	2 (100%)	9 (90%)
Rash	6 (26.1%)	4 (36.3%)	1 (50%)	1 (10%)
Grade 1	1 (4.3%)	1 (9.1%)	1 (50%)	0 (0%)
Grade 2	5 (21.7%)	4 (36.3%)	0 (0%)	1 (10%) **
Hypomagnesemia	4 (17.4%)	1 (9.1%)	0 (0%)	3 (30%)
Grade 1	3 (13%)	1 (9.1%)	0 (0%)	2 (20%)
Grade 2	1 (4.3%)	0 (0%)	0 (0%)	1 (10%) **
Hypomagnesemia with Rash	10 (43.5%)	5 (45.5%)	1 (50%)	4 (40%)
Grade 1	3 (13%)	2 (18.2%)	0 (0%)	1 (10%)
Grade 2	7 (30.4%)	3 (27.3%)	1 (50%)	3 (30%) **^, Δ^
Other (all Grade 5)	1 (4.3%) *	0 (0%)	0 (0%)	1 (10%) *^,^ **

* Multiple sclerosis flare thought to be driven by cetuximab. ** Led to therapy discontinuation, **^Δ^** in only 1 patient

## Data Availability

The data presented in this study are available on request from the corresponding authors. The data are not publicly available to protect the privacy of patients included in the study.

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
