# Peer review of "Cetuximab for Immunotherapy-Refractory/Ineligible Cutaneous Squamous Cell Carcinoma"

_cancers, 2023, doi:10.3390/cancers15123180_

Round 1
Reviewer 1 Report
The work presents a retrospective study on the use of ICI therapy in three cohorts to treat SCC, and the followup use of different strategies where ICI therapy has proven to be ineffective. The numbers are typical for a study of this type. The finding is interesting and is certainly worthy of further followup. The finding that Cetuximab can be used for tumours refractory to ICI is important. The work is clear and well written.
Author Response
We are grateful to the reviewer for their comment and the detailed analysis of our manuscript.
Reviewer 2 Report
- A brief summary: In the article by Marin-Acevedo et al., the authors have mentioned their single-institution retrospective experience on the use of cetuximab immediately following anti-PD1 therapy in patients with advanced cutaneous squamous cell carcinoma who fail or are ineligible for immunotherapy. According to their data, cetuximab exhibited rapid and durable responses. The results need to be validated in larger cohorts to establish the exact sequence of administration of either anti-EGFR therapeutics or ICI drugs, either alone or as a combinatorial strategy, in ACSCC patients, thereby prolonging their median survival with minimal side effects.
- General concept comments: In my opinion, the manuscript is vivid, relevant for the cancer therapeutics field, particularly addressing primary or secondary resistance in ACSCC patients. Although the article is scientifically sound, the major demerit of the study is the small sample size of patients selected for the study and the source of data being procured as a single institution retrospective design, which is well acknowledged by the authors themselves. This may result in unavoidable biased conclusions after the conclusion of the study. Furthermore, the side effects of cetuximab treatment should be addressed on a large scale before attempting anti-EGFR treatments post-ICI failure or as a combinatorial strategy. As an active cancer biologist, I am intrigued to learn more about the research on dosage of treatment strategies (either mono or dual therapy in cancer cells or tumor mouse models) affecting both the immune system and physiological processes in cancer patients to prolong their survival. Also, why was only one race selected for the study?
- Specific comments: The article is well written and grammatically correct; however, there is a minor spacing issue in the beginning of a sentence on lines 29, 69, 108, and 111 and the reason behind selecting only one race (white) of people for the study.
Author Response
General Concept Comments:
In my opinion, the manuscript is vivid, relevant for the cancer therapeutics field, particularly addressing primary or secondary resistance in ACSCC patients. Although the article is scientifically sound, the major demerit of the study is the small sample size of patients selected for the study and the source of data being procured as a single institution retrospective design, which is well acknowledged by the authors themselves. This may result in unavoidable biased conclusions after the conclusion of the study. Furthermore, the side effects of cetuximab treatment should be addressed on a large scale before attempting anti-EGFR treatments post-ICI failure or as a combinatorial strategy.
- As an active cancer biologist, I am intrigued to learn more about the research on dosage of treatment strategies (either mono or dual therapy in cancer cells or tumor mouse models) affecting both the immune system and physiological processes in cancer patients to prolong their survival.
We appreciate this insightful comment from the reviewer. In our study, all patients received the labeled doses of maintenance cetuximab while some did not receive the higher induction doses given concerns for toxicities. We agree that further investigation on dosage and treatment combination strategies are required. However, we felt this extended beyond the scope of our retrospective review in which wanted to focus on our clinical success using cetuximab in the modern era.
- Also, why was only one race selected for the study?
We thank the reviewer for such insightful question. We have included the following statement in the Discussion section (lines 346-349) as a limitation:
“The generalizability of our findings is limited by the fact that only white individuals were represented. Since we did not purposely exclude individuals of different races/ethnicities, this likely represents that CSCC is more common among white patients.”
Additionally, we cited the following source:
- Karia, P.S., J. Han, and C.D. Schmults, Cutaneous squamous cell carcinoma: estimated incidence of disease, nodal metastasis, and deaths from disease in the United States, 2012. J Am Acad Dermatol, 2013. 68(6): p. 957-66.
Specific comments:
The article is well written and grammatically correct
- There is a minor spacing issue in the beginning of a sentence on lines 29, 69, 108, and 111
We apologize for the oversight and thank the reviewer for this comment. The spacing issues have been corrected throughout the manuscript.
- The reason behind selecting only one race (white) of people for the study.
We thank the reviewer for such this pertinent question. We have included the following statement in the Discussion section (lines 346-349) as a limitation:
“The generalizability of our findings is limited by the fact that only white individuals were represented. Since we did not purposely exclude individuals of different races/ethnicities, this likely represents that CSCC is more common among white patients.”
Additionally, we cited the following source:
- Karia, P.S., J. Han, and C.D. Schmults, Cutaneous squamous cell carcinoma: estimated incidence of disease, nodal metastasis, and deaths from disease in the United States, 2012. J Am Acad Dermatol, 2013. 68(6): p. 957-66.
Reviewer 3 Report
Thank you for your interesting manuscript.
Major comments:
1. "Cetuximab immediately after ICI failure" is a unique idea, and it should be appraised.
2. The reason to select cetuximab is a little bit weak. Is it just from your experience?
3. One figure to show the design of the study would be helpful.
Minor comments:
1. Anti-EGFR therapies other than cetuximab or afatinib should be mentioned, if any.
2. Action mechanisms of this therapeutic strategy could be more explored.
Author Response
Major comments:
- "Cetuximab immediately after ICI failure" is a unique idea, and it should be appraised.
We thank the reviewer for this valuable comment. We believe that our findings are intriguing enough to warrant a prospective evaluation of our hypothesis within a larger cohort of patients. We also believe that tissue analysis prior to, during, and after anti-PD1/cetuximab therapy may shed light on the underlying mechanisms behind the enhanced activity using these therapies. We are planning on pursuing these analyses at our institution to further understand the interaction between cetuximab and ICIs.
- The reason to select cetuximab is a little bit weak. Is it just from your experience?
We thank the reviewer for this insightful comment. The selection of cetuximab derived from our encouraging anecdotal experience using this agent at our institution. Cetuximab is an option listed in the NCCN guidelines and potentially has a more favorable toxicity profile relative to chemotherapy, which is often not well tolerated in individuals of advanced age like our group of patients.
We have included the following statement in our Introduction section (lines 62-66), which reads as follows:
“It is thought to be better tolerated than chemotherapy, particularly in elderly patients which compromise the majority of cases of advanced CSCC.”
We included the following citation:
Bauman, J.E., K.D. Eaton, and R.G. Martins, Treatment of recurrent squamous cell carcinoma of the skin with cetuximab. Arch Dermatol, 2007. 143(7): p. 889-92.
We also mention the following in our Introduction section (lines 96-98):
“Our anecdotal experience using cetuximab following anti-PD1 failure was encouraging and became our preferred approach in the management of these patients.”
- One figure to show the design of the study would be helpful.
We thank the reviewer for this valuable recommendation. We have included a figure labeled as “Figure 1”.
In addition, the text has been modified under the Patients and Methods section (line 110-111) which now reads:
“The primary endpoint was ORR. Secondary endpoints included DCR, TTR, PFS, OS, and toxicity (Figure 1).”
Minor comments:
- Anti-EGFR therapies other than cetuximab or afatinib should be mentioned, if any.
We thank the reviewer for this suggestion. We included the following information into the discussion section (lines 328-330):
“Other anti-EGFR monoclonal antibodies (i.e., panitumumab) or tyrosine kinase inhibitors (i.e., gefitinib, erlotinib, afatinib, dacomitinib, osimertinib) could also be explored”.
- Action mechanisms of this therapeutic strategy could be more explored.
We thank the reviewer for this comment. We believe this is clearly the next step in investigation. As alluded to in our response for Question 1 above, we are aiming to collect paired biopsies of patients prior to, while on, and after immunotherapy use, including biopsies at the time of progression. We hope these samples will help us understand the mechanisms behind responses and will help us improve our sequencing strategies. Finally, we mention the following hypothesis in the Discussion section (lines 335-344) of manuscript:
“It is known that the EGFR signaling pathway modulates the tumor microenvironment and that anti-EGFR therapy directly affects responses to ICI. Based on this theory, one may speculate that the use of cetuximab prior to ICI could have negatively impacted responses to anti-PD1 and anti-CTLA-4 agents in our patients. Additionally, one may hypothesize that the reverse condition also stands and that ICI prior to cetuximab could modulate the tumor microenvironment and EGFR pathway, leading to improved responses to anti-EGFR therapy. Based on this theory and on the clinical results seen with other malignancies like melanoma, we believe the use of upfront ICI, when feasible, followed by targeted therapy at progression, could result in better outcomes compared to the opposite approach.”

Round 2
Reviewer 3 Report
Thank you for your proper revision of the manuscript.
This paper will hopefully inspire further research.